# Anterior Uveitis and Coats Disease in a 16-Year-Old Girl with Noonan Syndrome—A Case Report

**DOI:** 10.3390/children10101643

**Published:** 2023-09-30

**Authors:** Marta Świerczyńska, Agnieszka Tronina, Anna Lorenc, Erita Filipek

**Affiliations:** 1Department of Ophthalmology, Faculty of Medical Sciences in Katowice, Medical University of Silesia, 40-055 Katowice, Poland; 2Department of Ophthalmology, Kornel Gibiński University Clinical Center, Medical University of Silesia, 40-055 Katowice, Poland; 3Department of Pediatric Ophthalmology, Faculty of Medical Sciences in Katowice, Medical University of Silesia, 40-055 Katowice, Polandfilipek.erita@gmail.com (E.F.); 4Department of Pediatric Ophthalmology, Kornel Gibiński University Clinical Center, Medical University of Silesia, 40-055 Katowice, Poland

**Keywords:** Noonan syndrome, *PTPN11* mutation, anterior uveitis, ANA, Coats disease, retinopathy

## Abstract

*Background:* Noonan syndrome (NS) represents a fairly common genetic disorder with a highly variable phenotype. Its features include inherited heart defects, characteristic facial features, short stature, and mild retardation of motor skills. *Case presentation:* A 16-year-old Caucasian girl with NS reported visual deterioration, photophobia, and pain in the right eye (RE). The initial best-corrected visual acuity (BCVA) was 0.3 in the RE. An examination demonstrated conjunctival and ciliary body hyperemia, keratic precipitates, and flare in the anterior chamber. In addition, post-hemorrhagic floaters, tortuous vessels, and an epiretinal membrane in the RE were present. Diagnosis of unilateral anterior uveitis was made, and this resolved after the use of topical steroids and cycloplegic drops. Due to the presence of retinal telangiectasias and extraocular exudates (consistent with Coats’ disease (CD) stage 2A) in the RE, laser therapy was performed. The patient remains under constant follow-up, and after one year, the BCVA in the RE was 0.7. *Conclusions:* Here, we report the clinical characteristics, genetic findings, and retinal imaging results of a patient with NS. To our knowledge, this is, to date, the first report of an association of NS with a *PTPN11* mutation with anterior uveitis and CD.

## 1. Introduction

Noonan syndrome (NS) (OMIM 163950) is a mostly autosomal dominant inherited multisystem disorder with high heterogeneity in phenotype. The prevalence is estimated to be 1 in 1000–2500 live births [1,2,3]. Maternal transmission dominates in hereditary types, while de novo mutations tend to be paternal in origin [4,5]. The condition is caused mainly by gain-of-function (GOF) mutations in genes encoding proteins integral to the rat sarcoma/mitogen-activated protein kinase (RAS/MAPK) cell signaling pathway, which is crucial for cell cycle regulation, differentiation, growth, and apoptosis, as well as the inflammatory response [1,2,3]. Phenotypically overlapping diseases that result from germline mutations in genes implicated in the RAS/MAPK pathway are otherwise referred to as ‘RASopathies’ and, in addition to NS, include neurofibromatosis type 1, NS with multiple lentigines (previously named LEOPARD syndrome), capillary malformation–arteriovenous malformation syndrome, Costello syndrome, cardio–facial–cutaneous syndrome, and Legius syndrome [6]. NS diagnosis is established in progeny with evident findings and a pathogenic variant that is heterozygous in *PTPN11* (present in about 50% of patients), *BRAF*, *KRAS*, *MRAS*, *NRAS*, *RASA2*, *RRAS2*, *MAP2K1*, *RAF1*, *RIT1*, *SOS1*, and *SOS2* or as a biallelic or heterozygous mutant variant in *LZTR1* [1].

Both the complexity and severity of the disorder observed in patients with NS range from sparsely symptomatic adults reporting non-serious symptoms to neonates with life-threatening heart diseases. NS presents dysmorphic facial features (drooping eyelids, broadly set, downward-slanted eyes, low-set, posterior-turned ears, and a short neck with a low back hairline), which become significantly less distinct with age. In addition, patients with NS usually have a broad or band-shaped neck, an atypical chest profile with an upper pectus carinatum and lower pectus excavatum, short stature, multiple skeletal anomalies, congenital cardiac defects, hematologic diseases, lymphatic dysplasia, cryptorchidism, renal abnormalities, developmental delay of variable degree, hearing loss, and ophthalmological disorders [1,2,3].

International guidelines for the clinical management of NS recommend referral to an ophthalmologist at the time of diagnosis for a detailed evaluation of potential ocular manifestations such as refractive errors, amblyopia, strabismus, and nystagmus [1,2,7]. Similarly, an ophthalmological examination is strongly recommended in patients with suspected NS. Subsequent ophthalmological follow-ups should be performed approximately once every two years or more often if advised by a specialist.

## 2. Case Report

A 16-year-old Caucasian girl presented to the Ophthalmology Emergency Department due to photophobia and pain in the right eye (RE) for the last two days and deterioration of vision in this eye for about a week. She denied trauma, and her family history was unremarkable in terms of ophthalmic diseases. The patient was diagnosed with NS and mutation M504V (exon 13) in one allele of the gene *PTPN11* on chromosome 12. Patient was examined ophthalmologically only at the age of 5 after the diagnosis of NS, and no abnormalities were detected. The patient showed features of facial dysmorphia and short stature (Figure 1). She remained under the care of an outpatient nephrology clinic due to a duplicated pyelocaliceal system and renal failure.

Upon admission and examination, the best-corrected visual acuity (BCVA) was 0.3 in the RE and 1.0 in the left eye (LE). Intraocular pressure was within normal limits. Slit-lamp examination of the RE demonstrated conjunctival and ciliary hyperemia, keratic precipitates, posterior synechiae, and flare in the anterior chamber (1+), and insight into the fundus was hazy (Figure 2). In addition, post-hemorrhagic floaters, tortuous vessels, and an epiretinal membrane were present (Figure 3A), whereas the LE was normal (Figure 3B).

For the treatment of unilateral anterior uveitis, topical steroids and cycloplegic drops were used. Moreover, due to visible retinal telangiectasia and extrafoveal exudations in the superior temporal quadrant (consistent with stage 2A Coats disease (CD) according to the Shield classification system), 532-nm green laser photocoagulation was performed (Figure 4A,B).

The patient had regular follow-ups, anti-inflammatory treatment was gradually reduced, and no further laser therapy was required. Among the panel of tests performed to determine the etiology of the uveitis, only anti-nuclear antibody (ANA) at a titer of 1:1000 was present, and a further extended investigation did not confirm the presence of autoimmune diseases. After one year, the patient remained without new ocular findings, and the BCVA in the RE was 0.7 (Figure 4C).

## 3. Discussion

The ocular manifestations present in NS include refractive errors, ambylopia, strabismus, limited ocular motility, and nystagmus. Over 50% of patients have external ocular features (hypertelorism, ptosis, epicanthic folds, slanting palpebral fissures, and proptosis). Approximately two-thirds of patients with NS develop anterior eye segment abnormalities (prominent corneal nerves, corneal opacity, keratoconus, corneal pannus, microcornea, posterior embryotoxon, iris coloboma, persistent pupillary membrane, dilated episcleral vessels, uveitis, significant lens vacuolization, and cataracts). However, abnormalities of the posterior segment of the eye occur in about 20% of cases (excavation of the optic disc, druses, pitting, hypoplasia, atrophy, situs inversus, nerve myelination, and coloboma of the disc, retina, and/or choroid). Rare cases include retinopathy, tortuous vessels, angioid streaks, or retinal detachment [1,2,3,7,8,9,10].

In 1992, the first cohort study assessing the prevalence of ophthalmic disorders among people with NS revealed that 95% of participants had at least one ocular impairment [8]. However, another study from 2012 that included 35 NS patients showed at least one ophthalmic disorder in all of them and that 11% had three or more eye abnormalities [9]. However, a retrospective study considering 105 patients with NS resulted in the diagnosis of visual impairment due to binocular optic nerve abnormalities (atrophy or hypoplasia of the optic nerve) with associated nystagmus and/or strabismus in seven patients, mainly among patients with mutations in *RAF1*, *SHOC2*, and *KRAS* [10].

Although there are strong indications for ophthalmological examinations in all patients with and/or suspected of having NS, up to nine patients had never been to an ophthalmologist before, and a further three patients with reduced visual acuity and concomitant ophthalmological disorders were not fully evaluated until many years after the diagnosis of the underlying disease [10]. Significantly, NS is accompanied by a high prevalence of amblyogenic factors (strabismus, ptosis, astigmatism), which makes early ophthalmological examination crucial for maintaining normal visual acuity. However, some of the pathology may not appear until the later stages of life (e.g., keratoconus), and, therefore, the need for regular follow-up with an ophthalmologist is strongly supported. International guidelines for the clinical management of NS were issued in 2010 and 2013, which may explain why older patients were not immediately referred to an ophthalmologist after being diagnosed with NS. Another reason for late referral for eye examination may be the co-occurrence of other life-threatening disorders (e.g., cardiac abnormalities or pulmonary valve stenosis) that required more urgent management earlier in the child’s development [7,10].

Quaio et al. [11] demonstrated the coexistence of autoimmune diseases (celiac disease, autoimmune thyroiditis, hepatitis, vasculitis; antiphospholipid syndrome, systemic lupus erythematosus, and vitiligo) in 14% of NS patients with *PTPN11* mutations and an increase in autoimmune antibodies (most commonly, anti-nuclear antibodies (ANAs) and antithyroid antibodies) in 52% of those with RASopathies. Oliveira et al. [12] demonstrated an association of *NRAS* mutation and human autoimmune lymphoproliferative syndrome. In addition, Amoroso et al. [13] reported a case of a patient with NS who had as many as three concomitant autoimmune conditions—systemic lupus erythematosus, coeliac disease, and Hashimoto’s thyroiditis. In about half of these cases, NS was due to missense mutations in the PTPN11 gene, causing GOF for the non-receptor protein tyrosine phosphatase SHP-2 [14]. This enzyme is an intracellular pathway regulator that interacts with the adaptor proteins involved in B-cell function [15]. Furthermore, it maintains lymphocytes in a resting state and regulates the transcription of nuclear factor kappa-light-chain-enhancer of activated B cells (NF-kB), a factor involved in antibody production and activation of natural killer cells [16]. RAS, a GTP-binding protein, serves a major role in cellular mobilization, including in the inflammatory and proliferative responses required to maintain immune tolerance. Notably, immunological models have demonstrated an insufficiency in the RAS/MAPK signaling cascade that contributes to the development of autoimmunity [17,18]. Thus, we may assume that the predisposition to uveitis in patients with NS is genetically determined.

CD is an idiopathic, nonhereditary, progressive retinopathy characterized by telangiectatic and aneurysmal retinal vessels with intraretinal and/or subretinal exudations and fluid. Shields et al. [19] classified CD into five stages, and Daruich et al. [20] expanded stages 2B to include 2B1 and 2B2 according to the presence or lack of a subfoveal glial nodule (Table 1). The incidence of CD is 0.09 per 100,000 people. There is no ethnic or geographic correlation with its incidence. However, there is a strong predilection for the male sex (3:1) [21]. The most common prevalence is in the first and the second decades, and a younger age when the disease occurs is correlated with a heavier phenotype [22]. Prior studies have demonstrated that the condition typically presents with unilateral involvement and that only up to 10% of cases have CD present on both sides [23]. However, with the introduction of ultra-widefield imaging, the incidence of asymmetric bilateral CD is much higher than originally reported [24]. The exact pathophysiology of CD remains unclear, but it is postulated that changes in the endothelium of the retinal vessels and abnormal pericytes lead to a breakdown of the retinal blood barrier, while plasma leakage results in vessel wall thickening. Malformed pericytes and endothelial cells deteriorate, leading to the formation of aneurysms. Vascular closure causes ischemia and leaks lipid-rich exudates into the retina, with a subsequent increase in retinal thickness, formation of cysts, and retinal detachment [25,26].

The main goal is to obliterate anomalous vasculature and dilated aneurysms in order to inhibit the exudative process, for which a traditional green or yellow laser with a fairly extended duration (0.1–0.5 s) can be used [27]. At earlier stages of CD (1–3A), laser photocoagulation is indicated. Depending on the severity of the disease, multiple laser sessions are sometimes required. The interval between laser sessions should be at least three months, as the resolution of exudation occurs slowly [21]. Intravitreal injection of anti-vascular endothelial growth factor (VEGF) agents may be used as adjuvant therapy in CD, as they lead to the regression of abnormal vessels, reduction in macular oedema, exudation, and improvement or stabilization of visual acuity [28]. However, it should be noted that although VEGF levels are elevated in CD, they are not associated with its pathogenesis, and the administration of anti-VEGF injections does not ensure a cure for the underlying disorder [21]. In more advanced stages of CD, it is worth noting that cryotherapy and surgical treatment are indicated (Table 2).

SHP-2 has been demonstrated to exert significant effect on the activity of multiple growth factors and cytokine-dependent signaling pathways through its binding to growth factor receptors, cell surface adhesion molecules, and adaptor molecules [29]. SHP-2 mutations were found in NS in which vascular anomalies (arteriovenous malformations, aneurysms, hypoplasia of the posterior vessels, moyamoya) and childhood leukemias (juvenile myelomonocytic leukemia, B cell acute lymphoblastic leukemia, acute myeloid leukemia) may occur [30,31,32,33]. In addition, activating mutations in SHP-2 have been identified in sporadic solid tumors [34]. These mutations increase the migration of different kinds of cells, including endothelial cells, and enhance angiogenesis. Additionally, SHP-2 controls the intracellular pH of endothelial and vascular smooth muscle cells [29,32]. However, to date, only two cases of retinopathy have been described among patients with NS: unilateral exudative retinopathy—CD in a patient with *RAF1* mutation [10] and atypical rubella retinopathy [35]. To our knowledge, this is the first report of an association between NS with *PTPN11* mutation and with anterior uveitis and CD. We can speculate that the vulnerability to uveitis in patients with NS represents another genetically determined factor. However, given the relatively large number of patients with NS, it seems most likely that the coexistence of NS and CD is incidental.

## 4. Conclusions

Patients with NS have a wide spectrum of ocular abnormalities. Timely screening both in patients with diagnosed and in those with suspected NS is critical for detecting and then initiating the treatment of potentially vision-threatening abnormalities at the earliest possible stage. Prompt visual rehabilitation has an immense impact not only on the development of visual function, but also on the child’s overall development. In addition, regular follow-ups at an ophthalmology clinic play a crucial role in detecting changes that appear later in life.

## Figures and Tables

**Figure 1 children-10-01643-f001:**
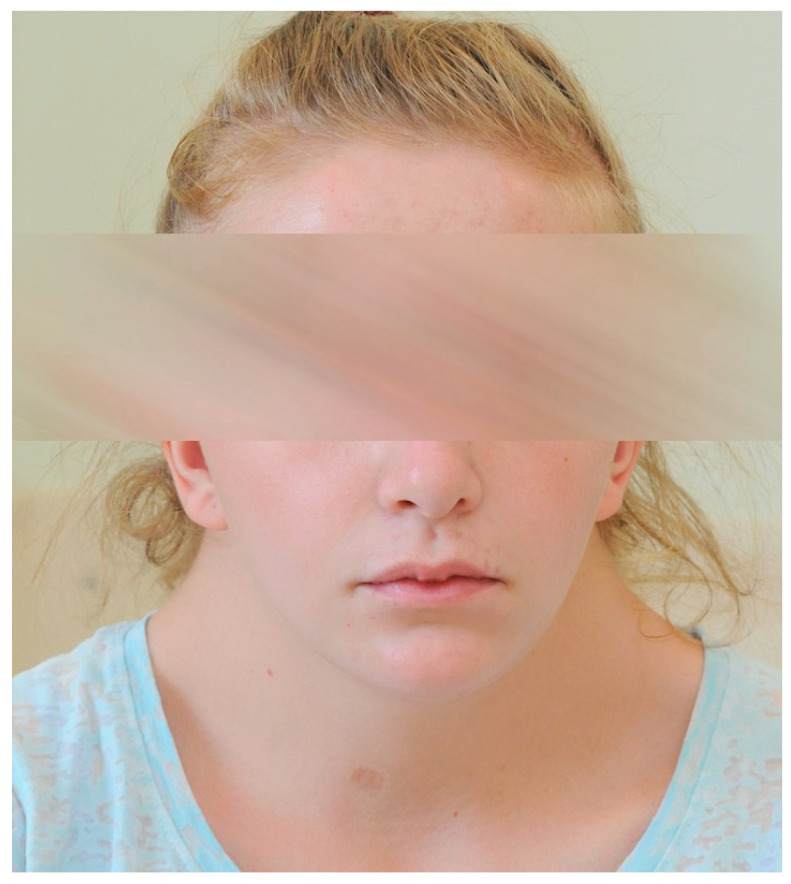
Photograph of the patient’s face; a triangle-shaped head, high anterior hairline, wide forehead, cupid bow appearance of upper lid, small chin, and neck skin webbing characteristic of NS are present.

**Figure 2 children-10-01643-f002:**
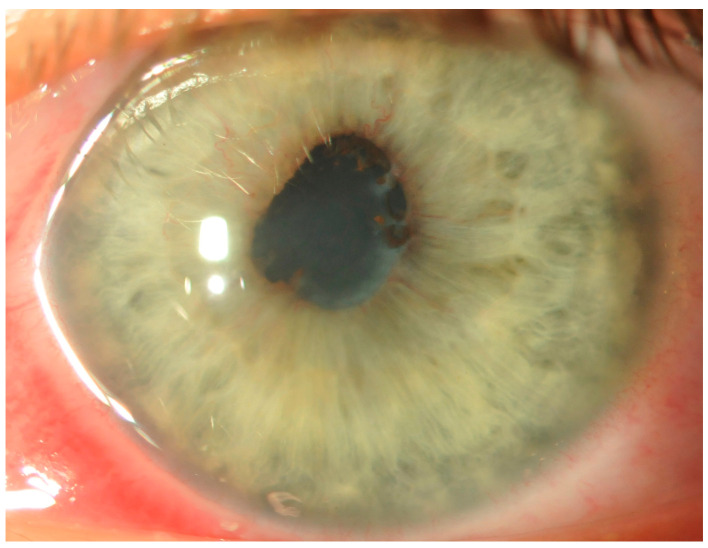
Photograph of the anterior segment of the RE; conjunctival and ciliary hyperemia, sparse keratic precipitates, and posterior synechiae are visible.

**Figure 3 children-10-01643-f003:**
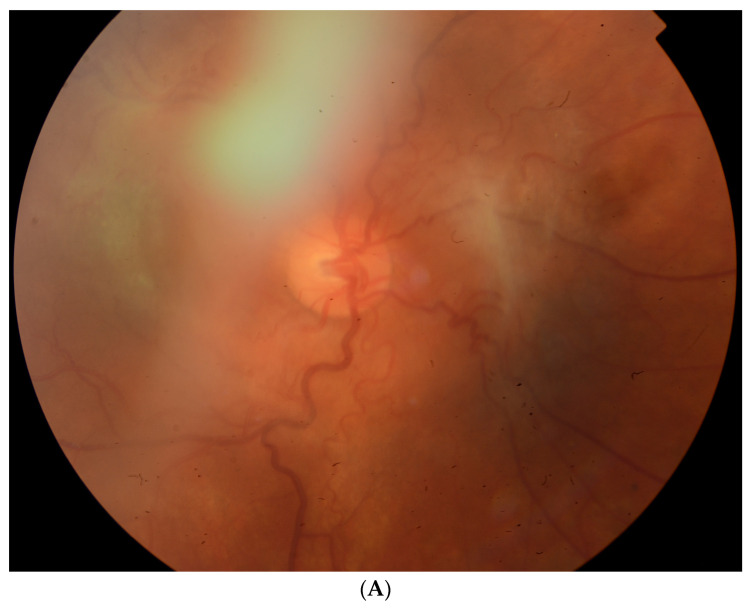
Photograph of the fundus (**A**) of the RE: The view is hazy, and post-hemorrhagic floaters, a tortuous course of vessels, and an epiretinal membrane are present. (**B**) The LE was within normal limits.

**Figure 4 children-10-01643-f004:**
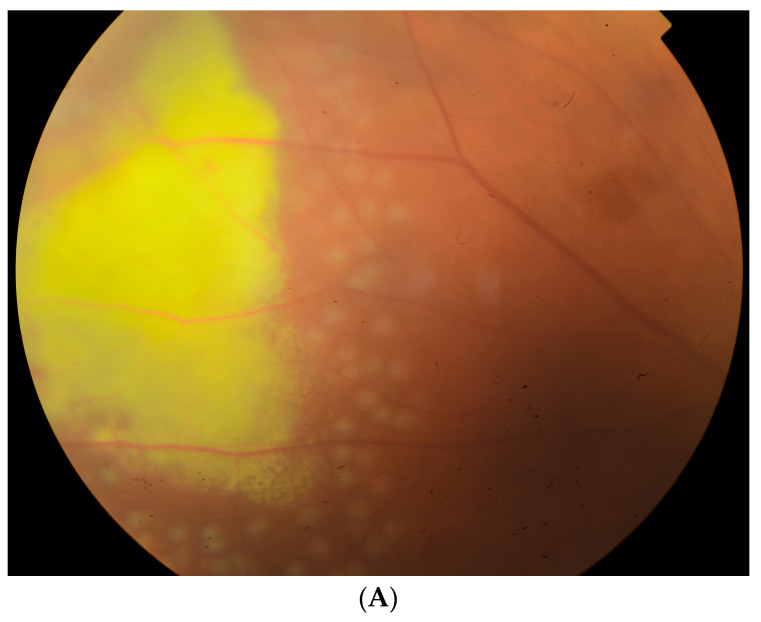
Photograph of the fundus of the RE (**A**,**B**) two days after laser therapy with retinal telangiectasia and extrafoveal exudations (consistent with stage 2A CD according to the Shield classification system) and (**C**) one year after laser therapy.

**Table 1 children-10-01643-t001:** Grading scheme for Coats’ disease [19,20].

Stage	Fundus Features
1	Retinal telangiectasia only
2	Teleangiectasia and exudation A. extrafoveal exudation B. foveal exudation 1. without subfoveal nodule 2. with subfoveal nodule
3	Exudative retinal detachment A. subtotal detachment 1. extrafoveal 2. foveal B. total retinal detachment
4	Total retinal detachment and glaucoma
5	Advanced end-stage disease

**Table 2 children-10-01643-t002:** Treatment modalities applied in CD in relation to the stage of the disease [21,28].

Stage		Treatment
1		Laser photocoagulation or cryotherapy
2		
3		Laser photocoagulation or cryotherapy; external drainage of total retinal detachment can be beneficial
4		External drainage of total retinal detachment; vitreoretinal surgery; glaucoma surgery may be necessary; occasionally, observation is advised
5	asymptomatic	Observation
	with painful eye	Enucleation
		Adjuvant therapy: intravitreal or periocular triamcinolone, anti-VEGF

## Data Availability

Not applicable.

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
