# Peer review of "Anterior Uveitis and Coats Disease in a 16-Year-Old Girl with Noonan Syndrome—A Case Report"

_children, 2023, doi:10.3390/children10101643_

Round 1
Reviewer 1 Report
Dear Authors,
This is an interesting case presentation about the association between Coat disease and Noonan syndrome. It is logically constructed and sets out the importance of ophthalmologic screening in all children diagnosed with this syndrome. Various ocular abnormalities in childhood, including low visual acuity of both eyes leading to visual impairment, are found in a large cohort of patients with Noonan syndrome. Coat disease is been described rare, mainly related with RAF1 mutation.
The available research is clearly presented and discussed, and the conclusion is supported by the evidence presented.
All the bibliographic data are recent and very well inserted in this article
Author Response
We would like to thank you for taking the time to review our manuscript and for your feedback.
Reviewer 2 Report
This is a case report of a 16-year-old girl with Noonan syndrome, anterior uveitis, and retinal telangiectasia.
Comments to the authors:
1. On page 1, lines 33-34, please add a reference for the sentence "Inherited cases have a predominance of maternal transmission, while de novo mutations are mainly of paternal origin"
2. Did you do a fluorescein angiography to confirm retinal telangiectasia? If so, please add a photo.
Author Response
We would like to thank you for taking the time to review our manuscript and for your feedback.
Response to Comment 1
We included the required literature.
Response to Comment 2
Fluorescein angiography was not performed in the patient due to the presence of
renal failure and the patient's parental lack of consent for the procedure.
Reviewer 3 Report
The authors report on a girl with NS (PTPN11 Mutation) with a seldom ocular manifestation.
The paper is well written, concise and clear. The authors also discuss hypotheses for the coincidence of both conditions and adress the limitation of this single case.
I have no comments.
Author Response

(The authors gave the same response as above.)
